# Understanding the Influence of 'Hot' Models in Climate Impact Studies: A Hydrological Perspective

Mehrad Rahimpour Asenjan[1], Francois Brissette[1], Jean-Luc Martel[1], Richard Arsenault[1]

[1]Hydrology, Climate and Climate Change Laboratory, École de technologie supérieure, Université du Québec, 1100 Notre-Dame Street West, Montreal, Quebec, Canada H3C 1K3

*Correspondence to*: Mehrad Rahimpour Asenjan (mehrad.rahimpour-asenjan.1@ens.etsmtl.ca)

**Abstract.** Efficient adaptation strategies to climate change require estimating future impacts and the uncertainty surrounding this estimation. Over- or under-estimating future uncertainty may lead to maladaptation. Hydrological impact studies typically use a top-down approach in which multiple climate models are used to assess the uncertainty related to climate model structure and climate sensitivity. Despite ongoing debate, impact modelers have typically embraced the concept of "model democracy" in which each climate model is considered equally fit. The newer CMIP6 simulations, with several models showing a climate sensitivity larger than that of CMIP5 and larger than the likely range based on past climate information and understanding of planetary physics, have reignited the model democracy debate. Some have suggested that hot models be removed from impact studies to avoid skewing impact results toward unlikely futures. Indeed, the inclusion of these models in impact studies carries a significant risk of overestimating the impact of climate change.

This large-sample study looks at the impact of removing hot models on the projections of future streamflow over 3,107 North American catchments. More precisely, the variability of future projections of mean, high, and low flows is evaluated using an ensemble of 19 CMIP6 GCMs, 5 of which are deemed "hot" based on their global equilibrium climate sensitivity (ECS). The results show that the reduced ensemble of 14 climate models provides streamflow projections with reduced future variability for Canada, Alaska, the Southwest US, and along the Pacific coast. Elsewhere, the reduced ensemble has either no impact or results in increased variability of future streamflow, indicating that global outlier climate models do not necessarily provide regional outlier projections of future impacts. These results emphasize the delicate nature of climate model selection, especially based on global fitness metrics that may not be appropriate for local and regional assessments.

## 1. Introduction

Understanding the impact of climate change on water resources and hydrology is crucial for developing effective strategies for mitigation and adaptation (Eyring et al., 2019; Miara et al., 2017). The output of hydrological (e.g. Karlsson et al. 2016), water quality (Prajapati et al., 2023) and sediment transport (Sabokruhie et al., 2021) impact assessment studies is dependent on the choice of the future climate change projections. Hydrologists primarily use climate projection outputs from GCMs (e.g. Tabari, 2020) to study these impacts. The Coupled Model Intercomparison Project (CMIP) provides standardized metadata from coordinated simulations by different climate modeling groups (Meehl et al., 2007). The more recent CMIP6 (Eyring et al., 2016) is gradually replacing the widely used CMIP5 from the last decade (Hirabayashi et al., 2021; Martel et al., 2022; Zhang et al., 2023).

The concept of "model democracy" has been widely used in impact studies (e.g. Collins et al., 2013; IPCC, 2014) despite criticism (Knutti, 2010). This approach considers climate simulations independent and equally plausible, and uses the ensemble mean and spread to define climate model uncertainty. Research has shown that the average of equally-weighted projections outperforms single models in simulating mean climatic patterns (Chen et al., 2017; Reichler & Kim, 2008). However, this approach may be less effective for CMIP6 ensemble as the validity of some simulations is under question (Hausfather et al., 2022).

The CMIP6 ensemble includes a subset of "hot models" that predict greater warming than previous predictions made by CMIP5 (e.g. Kreienkamp et al., 2020). These hot models have a climate sensitivity that exceeds the expected plausible range, which is based on observations and our understanding of planetary physics. They also exhibit a higher equilibrium climate sensitivity (ECS), a measure of the steady-state temperature increase in the event of doubled carbon dioxide ($CO_2$) concentrations in the atmosphere (Flynn & Mauritsen, 2020; Zelinka et al., 2020). The ECS values' range in CMIP6 models has increased to 1.8–5.6 °C compared to 2.1–4.7 °C in CMIP5, with an increase in multimodel mean of 3.9 °C in CMIP6 from 3.3 °C in CMIP5 (Zelinka et al., 2020).

However, a plethora of evidence based on observations and our understanding of planetary physics indicate that we can confidently restrict the likely range of future warming trend and, more importantly, give less weight to extreme estimates (Liang et al., 2020; Tokarska et al., 2020). Recently, more research has been focused on constraining the ECS based on historical and paleoclimatic data (Knutti et al., 2017; Sherwood et al., 2020) or emergent constraints (Cox et al., 2018; Nijsse et al., 2020; Shiogama, Watanabe, et al., 2022). For example, Sherwood et al. (2020) used multiple lines of evidence and concluded that the likely (with a 66% chance) ECS value is between 2.6°C and 4.1°C. Consequently, the most recent reports published by the Intergovernmental Panel on Climate Change (IPCC) have narrowed the likely ECS range to 2.5 and 4°C (IPCC, 2021). It should be noted that the uncertainty surrounding the cooling impact (both direct and indirect) of aerosols on

radiative forcing poses challenges in constraining future warming estimates (Bellouin et al., 2020; Forster et al., 2013; Smith et al., 2021). In essence, the current historical measurements do not provide a clear understanding of whether we are in a scenario of high sensitivity, fast-warming, accompanied by strong contemporary aerosol cooling, or if the situation is the opposite.

Climate change impact studies that include models with high ECS may be biased and may overestimate the magnitude of impacts (Hausfather et al., 2022). Using the full ensemble of CMIP6 projections without restricting the "hot models" may no longer be the most appropriate option for impact studies (Ribes et al., 2021). Incorporating climate models with high sensitivity into impact studies may potentially lead to an overestimation of the overall economic consequences arising from future climate changes (Shiogama, Takakura, et al., 2022). For instance, Shiogama et al. (2021) proposed a subset selection

method that involves screening out hot models as the first step. On the other hand, Palmer et al. (2022) found that models with higher sensitivity better represent some key climatic processes over Europe. While they were unable to provide robust physical explanations for their findings, it is worth noting that at the regional scale, hot models may provide valuable information that may be more important than the global warming trend for impact modelers, adding to the complexity of selecting models for regional impact studies.


The decision to weight climate models for impact studies remains controversial, but it is difficult to ignore the potential pitfalls of using hot models in these studies (Hausfather et al., 2022). This study aims to evaluate how including or excluding hot models in a multi-model ensemble affects the results of a large-scale hydrological climate change impact study. This influence is measured in terms of the magnitude and uncertainty of various streamflow metrics for 3107 North-American

catchments.

## 2. Materials and Methods

The data for this study was obtained from the HYSETS database, which contains hydrometeorological data from various sources for over 14,000 catchments in North America (Arsenault et al., 2020). The database includes all necessary data for the reference period of this study, including catchment boundaries (in the form of shapefiles), streamflow observations,

weather observations (from stations as well as multiple gridded and reanalysis datasets), and static catchment descriptors such as area, slope, elevation, land-use fractions, and soil properties. This study used the ERA5 reanalysis dataset for meteorological data, which was found to be a reliable alternative to gauge observations in a previous large-scale comparison study over the same study area (Tarek et al., 2020). To ensure representativeness, a subset of HYSETS catchments was selected using filters. First, catchments with drainage areas below 500 km$^2$ were excluded because daily hydrological models

would be inappropriate for modeling hydrological processes at smaller scales. Next, catchments required at least ten years of

data to ensure sufficient data for successfully calibrating hydrological models and bias-correcting climate models. Overall, 3107 catchments were retained.

Table 1 presents the list of 19 CMIP6 GCMs selected for this study. This list includes 5 hot models, defined by their ECS greater than 4.1. These models are: CanESM5 (ECS: 5.62), NESM3 (ECS: 4.68), IPSL-CM6A-LR (ECS: 4.52), EC-Earth3-veg (ECS: 4.3), EC-Earcth3 (ECS: 4.2). This study will be able to compare the uncertainty generated by the entire ensemble (19 models) to that of a reduced ensemble (14 models) obtained by removing the 5 hot models.

**Table 1. The 19 GCMs selected in this study and their corresponding ECS. ECS values were taken from either 1- Tokarska et al. (2020) or 2-Hausfather et al. (2022)**

| GCM | ECS |
| --- | --- |
| CANESM5 | 5.62[1] |
| NESM3 | 4.68[1] |
| IPSL-CM6A-LR | 4.52[1] |
| EC-Earth3-Veg | 4.3[1] |
| EC-Earth3 | 4.2[1] |
| ACCESS-ESM1-5 | 3.88[2] |
| GFDL-CM4_gr1 | 3.89[2] |
| GFDL-CM4_gr2 | 3.89[2] |
| MRI-ESM2-0 | 3.14[1] |
| MPI-ESM1-2-LR | 3.02[2] |
| BCC-CSM2-MR | 3.01[1] |
| MPI-ESM1-2-HR | 2.98[2] |
| FGOALS-g3 | 2.87[2] |
| GFDL-ESM4 | 2.62[1] |
| NorESM2-LM | 2.60[1] |
| MIROC6 | 2.57[1] |
| NorESM2-MM | 2.49[2] |
| INM-CM5-0 | 1.92[1] |

The impact study in this paper uses a traditional top-down hydroclimatic modeling chain consisting of one shared socioeconomic pathway (SSP8.5), 19 CMIP6 GCMs, one bias correction method, and one hydrological model. The study focuses solely on GCM uncertainty and doesn't consider other components, such as alternative SSPs, bias correction
methods, or hydrological models, which would add uncertainty to future projections. These have been explored in previous studies (e.g. Wilby and Harris, 2006; Chen et al., 2011; Giuntoli et al., 2018; Troin et al., 2022), and are outside the scope of this work. The reference period is based on the 1971-2000 time frame, while the future climate is based on 2070-2099.

Figure 1 illustrates the methodological framework for each study catchment. Precipitation and temperature data are first
extracted from 19 CMIP6 climate models under the SSP8.5 scenario for both the reference and future periods. Using precipitation and temperature from the ERA5 reanalysis over the reference period, climate data is then bias-corrected using the MBCn method. These bias-corrected climate scenarios are subsequently employed as inputs for a calibrated hydrological model to compute streamflows. These computed streamflows are then used to examine the impact of including (or not including) 'hot' models in the impact study, using a set of defined metrics. Further details are provided below.

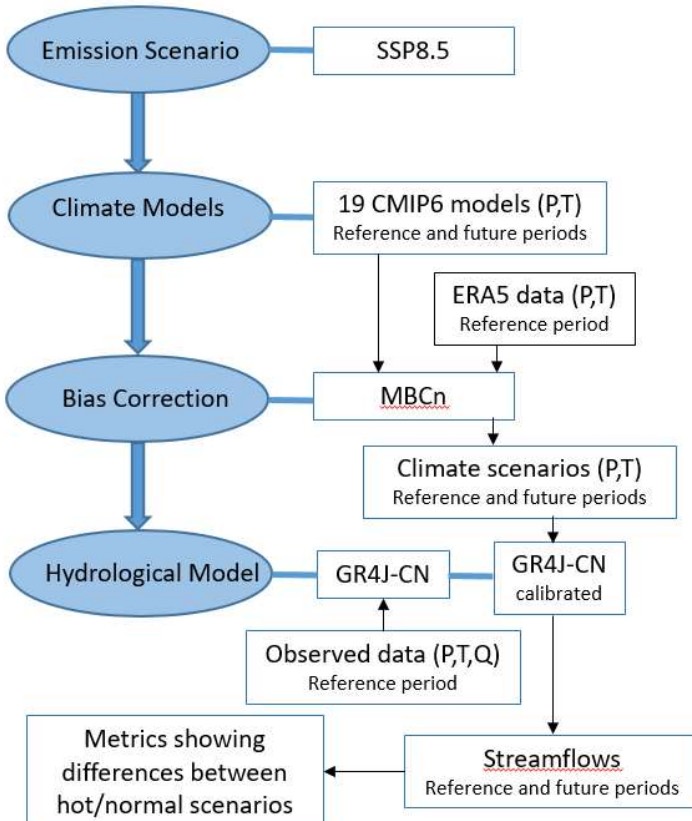


**Figure 1. Methodological framework performed for each of the study catchments.**

Climate models are mathematical representations of the Earth's climate system, based on current understanding of its physics
and chemistry. They are formulated using simplifying assumptions and parameterizations, but may not fully capture the complexity of the real climate system due to limited observations and understanding. As a result, climate models can be biased when compared to observations, due to factors such as model resolution, errors in reference datasets, and sensitivity to initial conditions. To ensure realistic impact simulations in impact studies, it is important to bias-correct climate model outputs. In this work, Cannon's (2018) N-dimensional multivariate bias correction (MBCn) method was used to correct
biases in daily precipitation and temperature. MBCn is considered the most advanced and efficient quantile-based multivariate bias correction method, as reported by studies such as Chen et al. (2018), Su et al. (2020), and Cannon et al. (2020). MBCn transfers the distribution of observational data to the corresponding distribution from the climate model while preserving its projection trends, crucial for climate change impact studies (Maraun, 2016). No downscaling was performed since this study was conducted at the catchment scale.

In this study, the GR4J lumped rainfall-runoff model (Perrin et al., 2003) was chosen to simulate streamflows. The model was selected due to the large number of catchments, which made it infeasible to use more complex, distributed models. Additionally, lumped models use averaged temperature and precipitation at the catchment scale, which is more consistent with the scale of GCMs, eliminating the need for downscaling. Lumped models have been shown to perform well in

simulating streamflows at catchment outlets (e.g. dos Santos et al., 2018; Reed et al., 2004). The GR4J model is simple, efficient, and high-performing compared to other lumped conceptual models. It uses precipitation, potential evapotranspiration (PET), and catchment surface area as inputs. To account for snow accumulation in some catchments, the GR4J model is linked with the CemaNeige snow module (Valéry et al., 2014), resulting in a 6-parameter model (GR4J_CN). The GR4J_CN model combination has been used in many studies, including climate change impact studies, and has been

shown to perform well under a wide range of conditions (e.g. Riboust et al., 2019; Tarek et al., 2020; Wang et al., 2019). The calibration was performed using the Kling-Gupta Efficiency (KGE) metric. The KGE metric (Gupta et al., 2009) directly combines the bias, ratio of variance, and correlation into a single metric. It provides a more robust and refined assessment of model performance when calibrating hydrological models, addressing the drawbacks of the Nash-Sutcliffe Efficiency metric (NSE, Nash & Sutcliffe, 1970) (Knoben et al., 2019). Figure 2 presents the location of the 3107 retained catchments, each

having a KGE calibration value above 0.5.

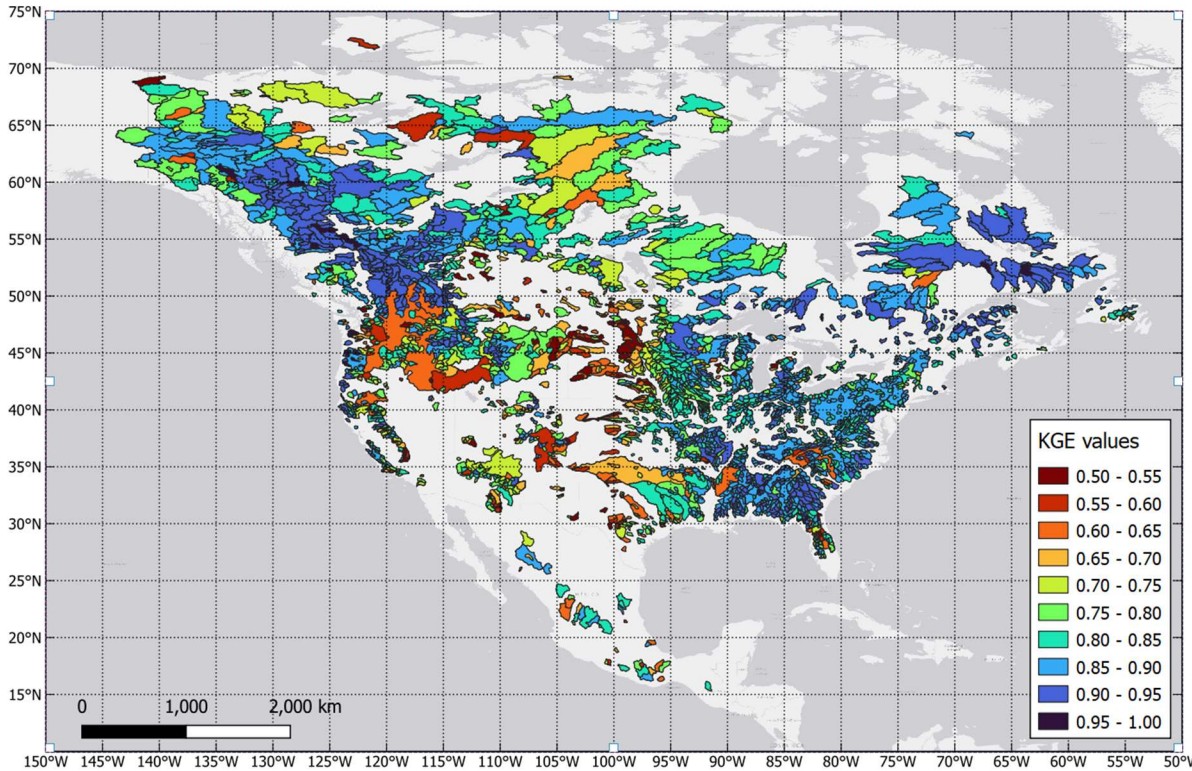

**Figure 2. Study catchment location. The color scale corresponds to the hydrological model KGE calibration score over the reference period. Only catchments with available data, KGE values higher than 0.5 and area larger than 500 km² were selected.**

The hydroclimatic modeling chain described above generated 19 different 30-year time series of daily streamflow for the 2070-2099 future period, each corresponding to one of the 19 GCMs listed in Table 1. Three streamflow metrics were extracted from each 30-year time series, representing mass balance ($Q_{mean}$) and high ($Q_{max}$) and low ($Q_{min}$) flows:

- $Q_{mean}$: obtained by averaging daily streamflow over the 30-year period.

- $Q_{max}$: obtained by averaging the 30 annual maximum simulated streamflows.

- $Q_{min}$: obtained by averaging the 30 annual minimum simulated streamflows.

These metrics will be used to assess the impact of removing hot climate models across a range of flow conditions.

Figure 3 presents the three dispersion metrics used in this study to compare the spread (or uncertainty) of future projections of streamflow metrics. For the three streamflow metrics, 19 values from the original ensemble and 14 from the reduced ensemble for both the reference and future periods are extracted. The spread of the streamflow projections over the reference

period is small, but it is not zero due to imperfect bias correction and the hydrology model's strong non-linear response to precipitation and temperature inputs. The spread is comparatively much larger in the future period, mainly due to differences

in sensitivity and structure of the climate models.

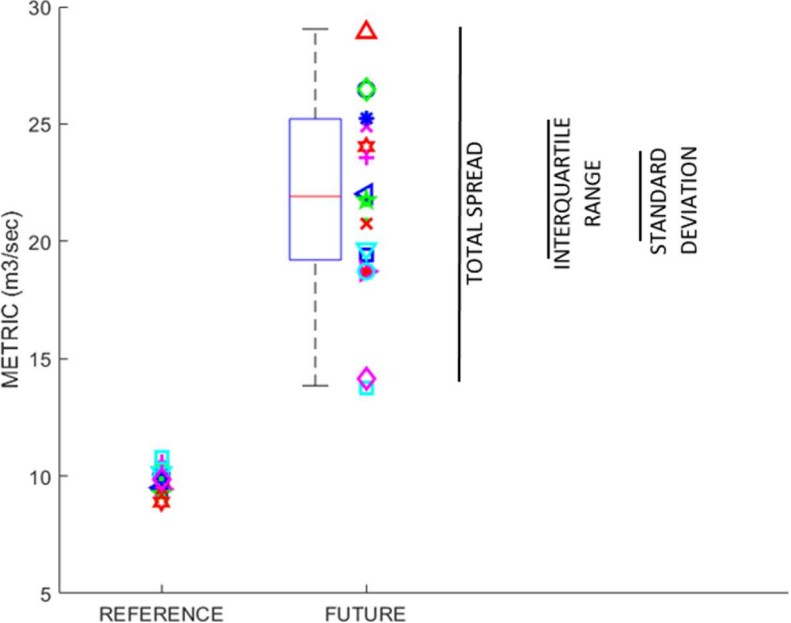

**Figure 3. Representation of the dispersion metrics used in this paper. Each marker represents one of the 19 climate models. METRIC will either be Q<sub>mean,</sub> Q<sub>max</sub> or Q<sub>min</sub>, all having units of m³/sec.**

Total spread (TS) is defined as the full range of future streamflow responses:

$$TS = metric_{max} - metric_{min}$$

The interquartile range (IQR) is defined as the distance between the 75th and 25th quantiles of the distribution as shown by the blue rectangle in the boxplot in Figure 3.

175                                                         $$IQR = Q_{75} - Q_{25}$$

Finally, the standard deviation (σ) is the standard mathematical measure of dispersion. In the case of a normal distribution, the standard deviation and interquartile range are perfectly correlated, but this may not be the case for a skewed distribution.

All three metrics have units of m³/s and are therefore dependent on catchment size and, to a lesser extent, mean annual precipitation. To account for this, the metrics will be presented in a non-dimensional form:

$$TS_{nd} = \frac{TS_{14}}{TS_{19}}$$

Where $TS_{19}$ and $TS_{14}$ respectively represent the total spread for the full and reduced ensemble. $TS_{nd}$ varies between 0 and 1, with $TS_{nd}=1$ meaning that no reduction in total spread was obtained by removing the five warm models from the ensemble,

and $TS_{nd}=0$ signifies that the total spread of the reduced ensemble has been totally eliminated.

Similarly, for the interquartile range ratio, we find:

$$IQR_{nd} = \frac{IQR_{14}}{IQR_{19}}$$

However, in this case, the potential values vary in the 0 to $\infty$ range. More practically, a value below 1 indicates that the IQR

has been reduced by removing the five hot models from the ensemble, whereas a value larger than 1 shows the opposite. The latter is possible if the removed models are somewhat close to the median of the ensemble.

Finally, for the standard deviation the following ratio is used:

$$\sigma_{nd} = \frac{\sigma_{14}}{\sigma_{19}}$$

where a value below 1 indicates a smaller standard deviation for the reduced ensemble, and the opposite for a value above 1. $\sigma_{nd}$ has the same possible range of values as $IQR_{nd}$ (0 to $\infty$).

## 3. Results

Figure 4-a presents the box plots of projected temperature increases for each of the 3107 catchments and for each climate model. The box plots provide a visual representation of key elements of the temperature increase distribution. The median of

the distribution is shown as the red line near the centre of the blue rectangle, which delimits the interquartile range (Q75 and Q25 for the upper and lower end of the rectangle). The whiskers represent the 2.5th and 97.5th quantile of the distribution, providing a 95% coverage of the dataset. Quantiles below 2.5 and above 97.5 are shown as dots. Results indicate that the distribution of projected temperature increases generally follows the same order as the ECS values presented in Table 1. However, there are some differences, which are not unexpected as global-scale ECS values are compared to regional-scale

$\Delta T$ values. The five hot models are ranked as the first, second, third, fifth, and sixth hottest regional models based on median values (considering that GFDL-CMA gr1 and gr2, respectively fourth and fifth, are actually the same model with different spatial resolutions).

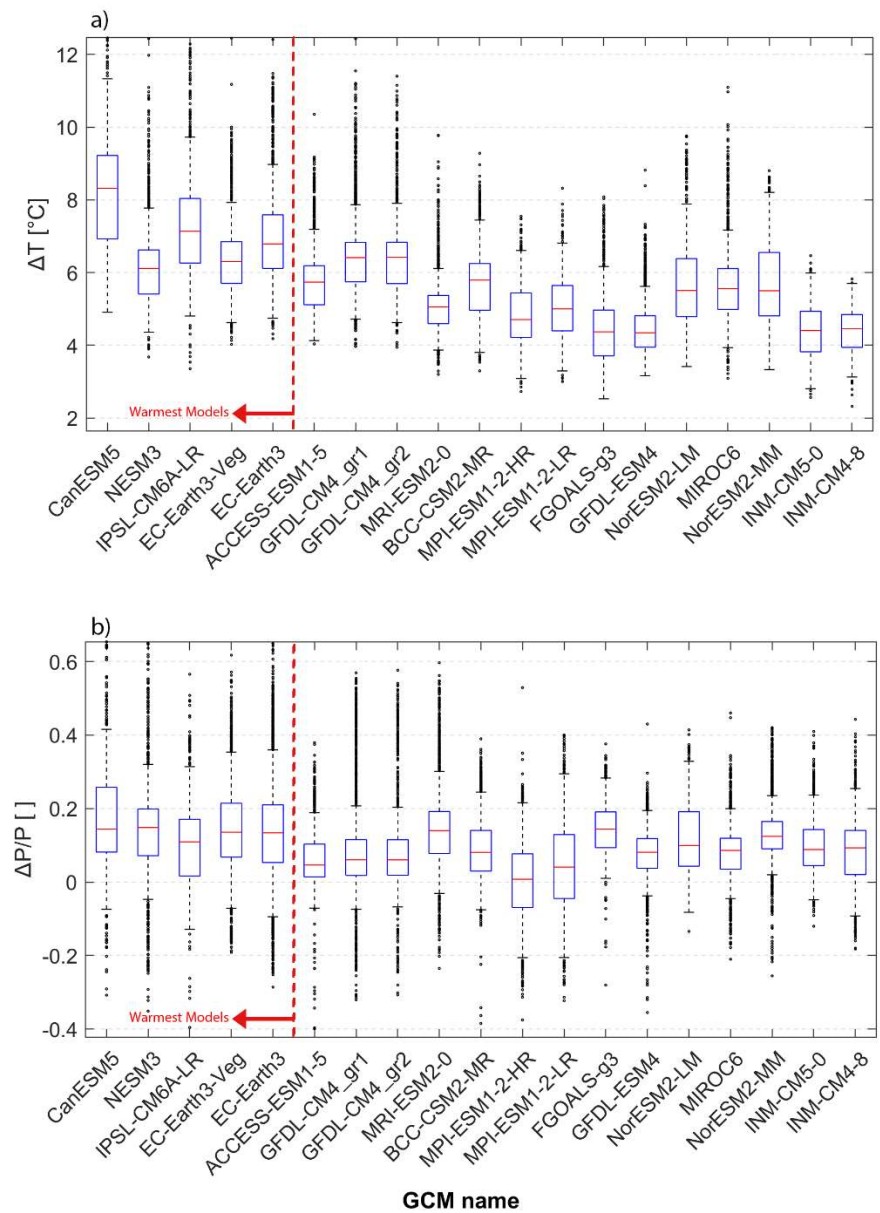


**Figure 4. a) Distribution of projected temperature increase (ΔT) and b) projected relative annual precipitation increase (ΔP/P) for the 19 CMIP6 selected model for the 2070-2099 future period, compared to the 1971-2000 reference period. Each boxplot represents the distribution of projected increases for the 3107 study catchments. The climate models are ordered in terms of their global-scale ECS values, starting with the largest to the left. The boxplot whiskers correspond to the 2.5th and 97.5th quantiles and a few catchment that were beyond the Y-axis limits are not shown.**


Figure 4-b presents the boxplots of the projected changes in relative precipitation between the future and reference periods

($\frac{P_{fut}-P_{ref}}{P_{ref}}$). The boxplots depict the distribution of the projected precipitation changes for each of the 3107 catchments. Results indicate that the hot models, identified by their ECS values, are also among the models with the largest projected changes in relative precipitation. Specifically, the five hot models are all within the group of the eight wettest models. The models with more modest increases in precipitation (e.g., MPI-ESM, ACCESS) are also among the cooler models. This trend is expected, as a warmer atmosphere can hold more moisture (up to 7% per °C, according to the Clausius-Clapeyron

relationship), leading to more precipitation. Increased precipitation may mitigate the anticipated impacts of warmer models, such as increased evapotranspiration.

In order to show regional patterns related to Figure 4, Figure 5 displays the mean ΔT (4a) and mean ΔP/P (4b) ratios between hot models and normal models. For temperature a red color indicates that hot models are warmer than the other models on

average. For precipitation, blue colors highlight increased precipitation in the hot models compared to the normal models. Overall, the hot global models exhibit a systematically larger temperature increase over the entire study domain. The hot models mostly exhibit increased precipitation compared to the normal models. However, the west coast of the U.S., as well as some catchments in the southwestern U.S., exhibit a decrease in precipitation according to the hot models. These observations underscore the regional variability in temperature and precipitation patterns when comparing hot and normal

models.

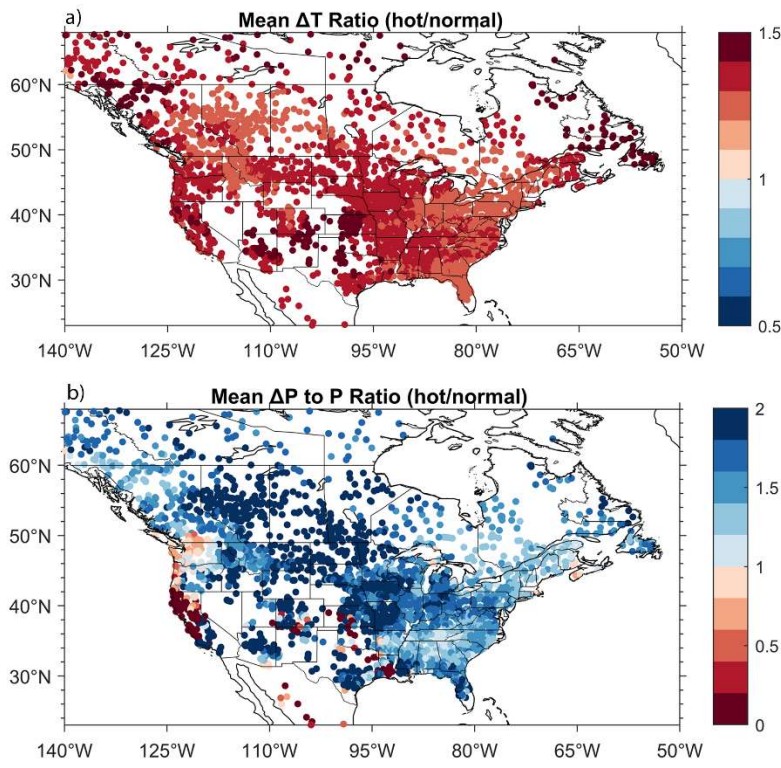

**Figure 5. Mean ΔT (a) and ΔP/P (b) ratios (hot models to normal models). For ΔT, a red color indicates that hot models, on average, are warmer than their normal (non-hot) counterparts. For ΔP/P, a blue color shows that hot models are wetter than their normal (non-hot) counterparts. The graphs represent the differences computed between the future and reference periods.**

Figure 6 presents the ratio of mean projected streamflow changes (hot models/normal models) for $Q_{mean}$, $Q_{max}$ and $Q_{min}$. A blue color indicates larger projected streamflows by the 'hot' models. Results show spatial patterns which differ depending on the streamflow metrics. Hot models project higher mean flows over most of the study domain, except in the south-west regions, where increased evapotranspiration nullifies potential increases in precipitation. For $Q_{max}$, increases are mostly localized in the Eastern US, whereas $Q_{min}$ are widely increasing in Canada and mostly decreasing in the US.

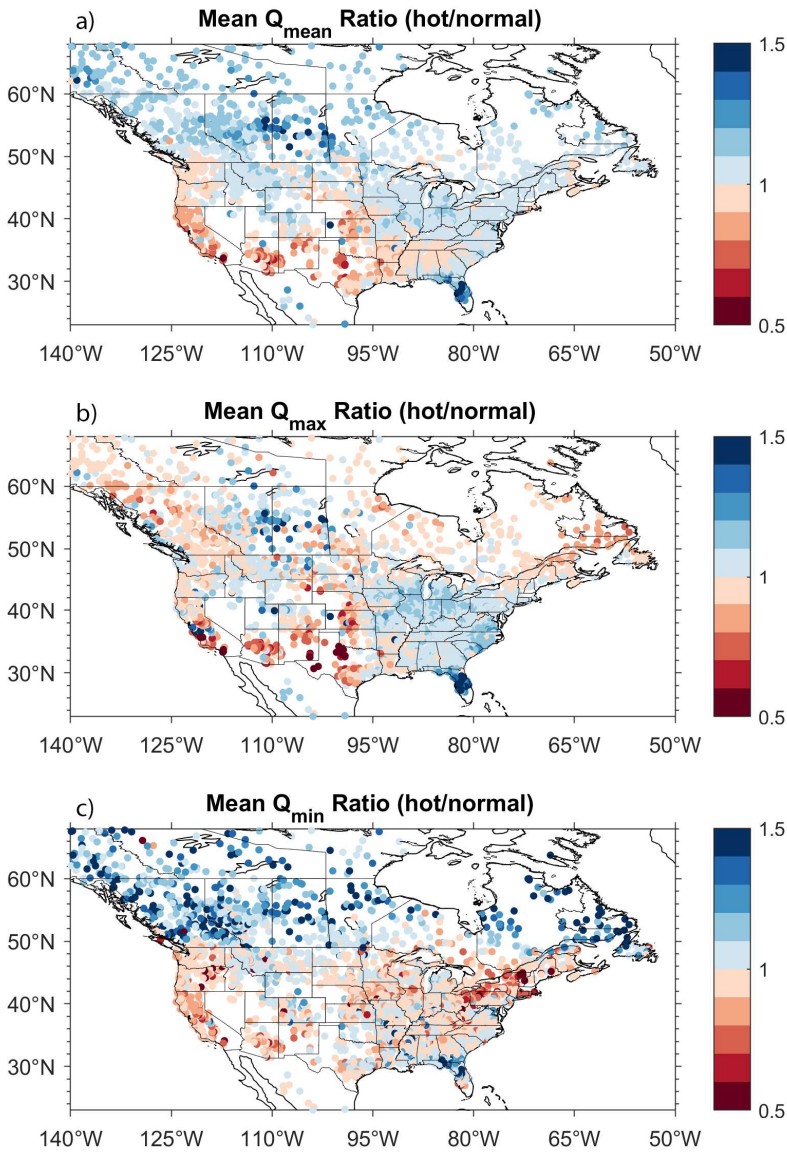


**Figure 6: Ratio of mean projected changes: 'hot' divided by normal models.  a): Qmean; b) Qmin; c): Qmax). A blue color shows that hot project larger streamflows than their normal (non-hot) counterparts.**

Figure 7 presents the $TS_{nd}$ for mean (Q_mean), annual max (Q_max), and min (Q_min) streamflow obtained by removing the 5 hot models from the 19-member ensemble. A dark red color indicates no reduction in TS with the reduced ensemble, while lighter colors indicate a reduction. It can be seen that there is a clear spatial pattern that is relatively similar for all three streamflow metrics. The largest reductions in TS are seen in the northern regions as well as in the US southeast, and along

the US Pacific coast for $Q_{mean}$ and $Q_{min}$. For all other regions of the US, no reduction in TS is observed. The reduced spread observed in the northern regions is smaller for $Q_{max}$. Despite these trends, a lot of variability remains present, with neighbouring catchments sometimes showing contrasting behaviour. More specifically, 57.0% of the catchments see a decrease in TS for $Q_{mean}$, 53.3% for $Q_{max}$ and 61.7% for $Q_{min}$.

The data from Figure 7 are shown in the form of boxplots in the left side of each panel to better illustrate the range of TS reduction. It shows that the median $TS_{nd}$ is relatively high for all three streamflow metrics: $Q_{mean}$ (0.96), $Q_{max}$ (0.95) and $Q_{min}$ (0.93). This is primarily because a significant number of catchments see no reduction in TS (43%, 46.7%, and 38.3% respectively). However, there is a significant reduction in TS observed in many catchments, and this decrease is strongly dependent on the geographical location of the catchments. Additionally, it can be seen that removing the hot models has a greater impact on $Q_{min}$ than on the other two metrics.


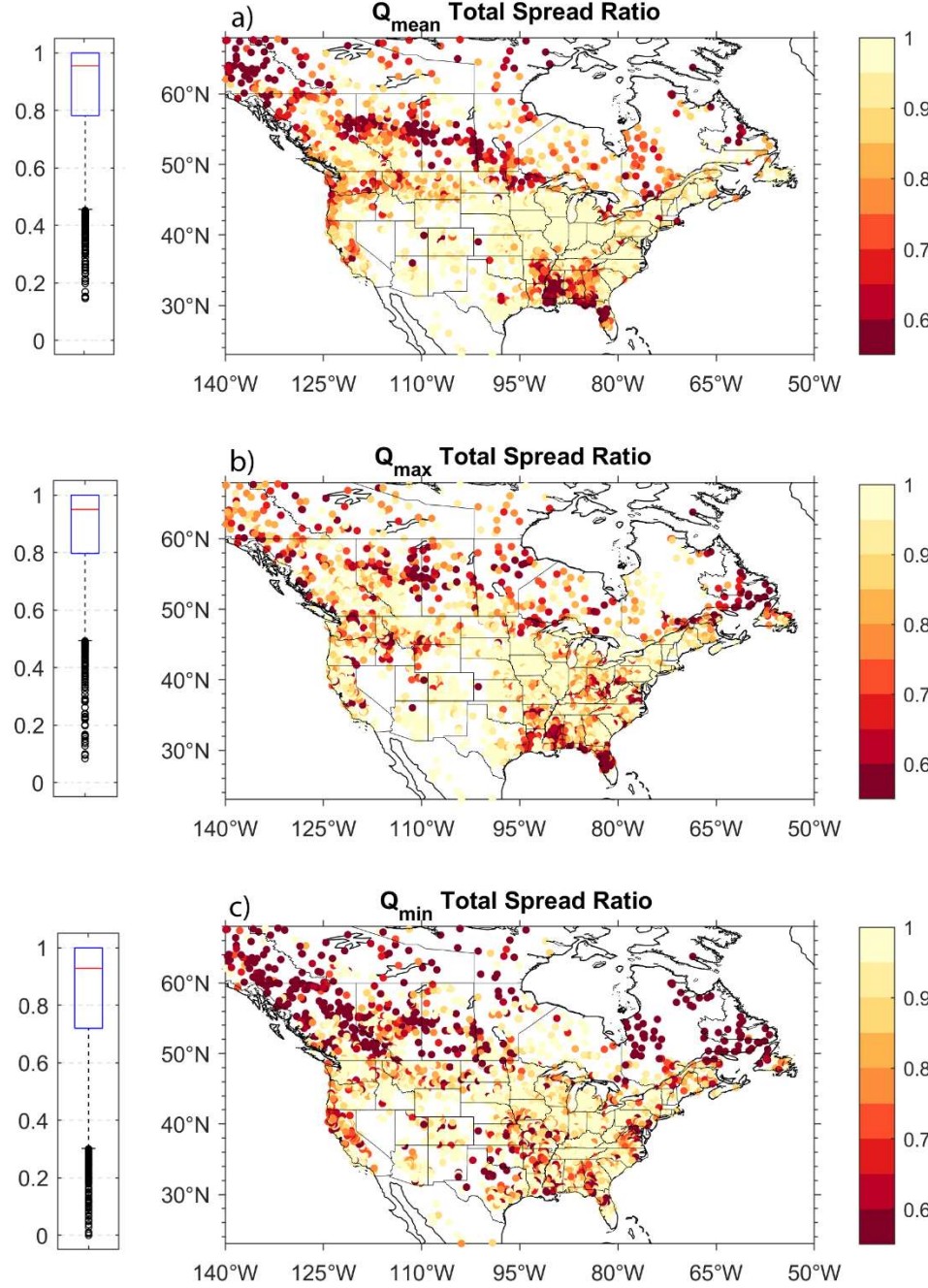

**Figure 7. Total spread ratio ($TS_{nd} = \frac{TS_{14}}{TS_{19}}$) for Q_mean (a), Q_max (b), and Q_min (c) resulting from the removal of the five hot models. Boxplots are shown in the left.**


The $TS_{nd}$ is heavily impacted by outliers and may not accurately represent the overall spread of models. Figure 8 presents the $\sigma_{nd}$ for the three streamflow metrics. A red color ( $\sigma_{nd}>$ 1) indicates that the model spread has increased following the removal of the hot models whereas a blue color ( $\sigma_{nd}<$ 1) corresponds to a decrease. Results indicate that removing the hot models consistently reduces $\sigma_{nd}$ in Canada for $Q_{mean}$ and $Q_{min}$, and to a lesser extent for $Q_{max}$. However, in CONUS, the

results are more complex with a lot of regional variability. Removing outlier models in the north central, north-east, and southwest of the US results in an increase in $\sigma_{nd}$ for both $Q_{mean}$ and $Q_{max}$. Overall, as shown in the boxplots of Figure 8, removing the hot models likely reduces the spread in roughly two-thirds of catchments, while one-third see an increase. These values are larger than those obtained for TS. The Trends seen in $IQR_{nd}$ is also very similar to that of $\sigma_{nd}$ (see figures S1 and S2).


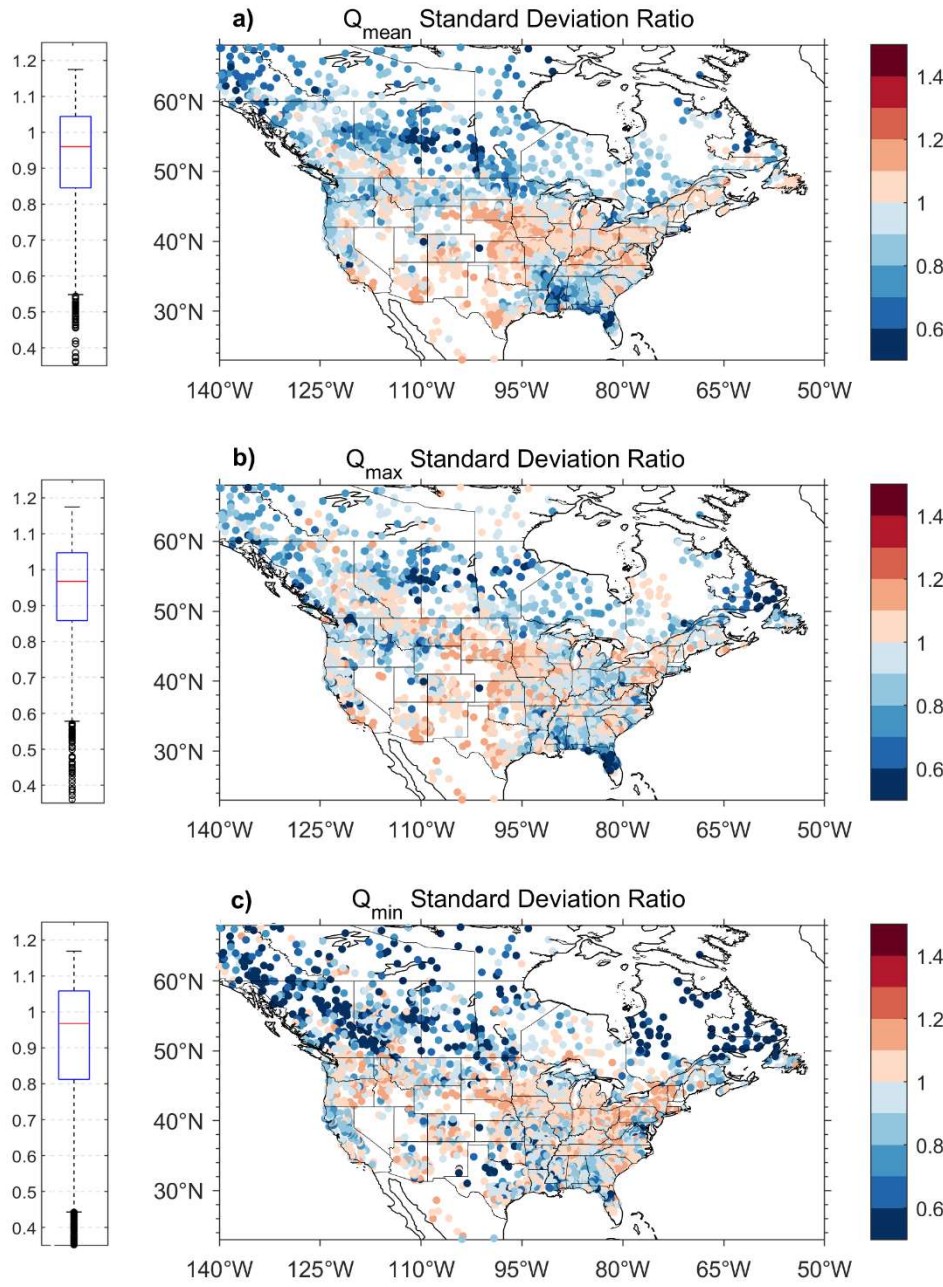

**Figure 8. Standard deviation ratio ($\sigma_{nd} = \frac{\sigma_{14}}{\sigma_{19}}$) for Q$_{mean}$ (a), Q$_{max}$ (b), and Q$_{min}$ (c) resulting from the removal of the five hot models. Boxplots are shown in the left side of each panel.**

## 4. Discussion

Uncertainty is a key factor in assessing the impact of climate change. Different models and techniques, including various climate models, can lead to diverse climate projections and scenarios. Climate change interacts with other stressors, such as land use change and population growth, in complex and unpredictable ways, making it important to accurately address uncertainty in climate impact studies to develop effective adaptation measures. Incorrectly representing uncertainty can lead to poor adaptation.

With the increased future temperatures, an intensification in the hydrological cycle is expected. However, it does not guarantee an automatic increase in water flow rates. This is because the rise in average temperature can also have a considerable impact on evapotranspiration. The outcome of these two factors working together is complex and varies based on the geographical location and primary climate zones. The research paper indicates that regions characterized as 'hot' tend to be associated with increased precipitation, further complicating the relationship between temperature and water flow.

Results show that removing the "hot models" is likely to reduce the spread of three streamflow metrics. Between 60% and 75% of catchments show a decrease in the spread of future streamflow projections, indicating that the hot models are outliers or further from the mean than the average model. In such cases, keeping the hot models would result in an overestimation of future streamflow uncertainty. However, removing the hot models also led to an increase in the spread in certain regions, indicating overconfidence in the results. This means that while the hot models are outliers with respect to ECS, they may not be outliers with respect to impact studies. Generally, a reduction in spread was evident in northern regions such as Canada and Alaska, as well as the coast of California and the southeastern region of the US. Shiogama, Watanabe, et al., (2022) also concluded that the inclusion of hot models leads to an overestimation of annual mean precipitation increases in Alaska, Canada, and the western United States, where there is a substantial decrease in the variability of streamflow metrics.

A reduction in the spread of future streamflow is expected when removing the hot models or reducing the number of climate models. A bootstrap methodology was used to determine if the changes in spread were due to a reduction in the number of models. This was conducted by selecting a random sample of 14 (out of 19) models 100 times and computing the average standard deviation ratio. This was repeated for all catchments and the aggregated results are shown in Figure 9.

The results indicate that removing five random models results in a decrease in the standard deviation ratio almost 75% of the time for all three streamflow metrics, but the median spread reduction ratio for this spread metric is extremely small (about 0.99 for all three streamflow metrics). This shows that removing the 5 hot models has a much larger impact than removing 5 random models. The spread reduction observed in many catchments is therefore not solely related to a reduction in the number of models.

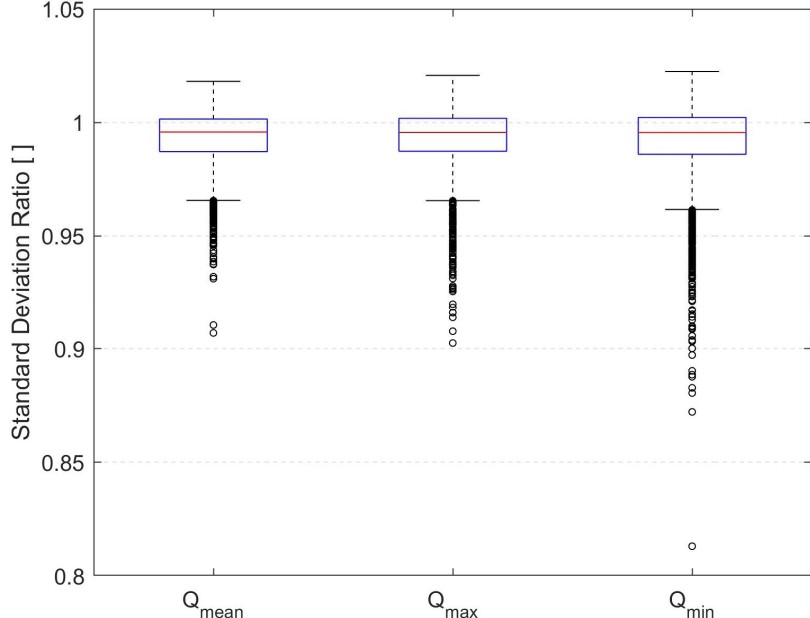

**Figure 9. Boxplots of the average standard deviation ratio for Q$_{mean}$, Q$_{max}$, and Q$_{min}$ resulting from the removal of 5 random models, after sampling 100 random combinations of 5 models.**

At first glance, there is a strong physical reasoning for removing climate models with equilibrium climate sensitivity (ECS) exceeding values expected from current data and understanding of planetary physics (Ribes et al., 2021; Shiogama et al., 2021). However, it should be noted that most impact studies are conducted at the regional or local scale and these models may not be considered outliers at these scales. This study found that while globally hot models may still be among the hottest in the study domain, they are not consistently the hottest, raising questions about whether their global behavior should automatically eliminate them from regional studies.

In this study, the climate performance of these models (such as their ability to represent climatic, hydroclimatic, or hydrological metrics) was not evaluated. The goal was to examine the impact of removing 5 hot models from a 19-member ensemble. However, it is important to note that judging climate models based solely on their ECS values may result in the removal of models that have desirable characteristics at the regional scale (e.g. Palmer et al., 2022). Additionally, keeping hot models may also be useful from an impact perspective as they may provide a clearer picture of future changes, as internal variability is less likely to obscure changes. This is similar to the rationale behind using high-emission scenarios in impact studies, such as SSP8.5, even though they may not be considered realistic scenarios anymore (e.g. Hausfather and Peters, 2020). It is important to consider worst-case scenarios when analysing potential outcomes, as high levels of greenhouse gas

emissions, or high model sensitivity, such as those projected in SSP8.5 or high ECS models, are not unrealistic, even though they may be less likely. While it is valuable to consider these high-end scenarios, it should be made clear that they are indeed worst-case scenarios.

In this study, the question of whether to remove the "hot models" for impact studies is complex. Results showed that for about one-third of all catchments, removing these models increased the future uncertainty of streamflow. This suggests that these "hot outliers" may not always be "hydrological outliers" when put through a hydrological modeling process. Hydrological models are well-known for being highly non-linear integrators of weather variables such as temperature and precipitation, and these results align with findings from other studies that have demonstrated the complex relationship between climate model projections and hydrological projections (e.g. Chen et al., 2016; Ross and Najjar, 2019). The fact that the CMIP6 hot climate models tend to be wet models may also be a factor in these results, as increased evapotranspiration could be offset by increased precipitation, leading to somewhat average results for the wrong reasons.

The regional impact of model importance is also compared (see figures S3 and S4 supporting information), which demonstrate the total spread ratio resulting from removing a single climate model and creating an 18-member ensemble. CanESM5 (Figure S3) and NESM3 (Figure S4) have the highest global sensitivity in this study. Removing CanESM5 leads to a clear reduction of total spread in Alaska and Yukon (for $Q_{mean}$ and $Q_{min}$) and in the Southeast USA for $Q_{max}$, indicating that CanESM5 is an outlier in these regions. Conversely, removing NESM3 does not result in significant decreases in spread over most of the study domain, as the high ECS value of NESM3 does not automatically translate into a correspondingly higher level of regional warming (see also fig. 4), demonstrating that it is not an outlier in most regions. This underscores the strong regional differences among globally identified hot models.

The only uncertainty in this study is that originating from GCM/EMSs. As stated earlier, in most impact studies, additional sources of uncertainty would also be incorporated. Additional greenhouse gases emissions scenarios would be selected as well as other impact models (e.g. hydrology models). Downscaling and additional bias correction may be performed. These additional components are likely to generate additional uncertainty which may, in some cases, dwarf that of climate models. As such, many of the differences observed in this paper between the original and reduced climate model ensembles may have little impact on the final uncertainty estimation. For example, for low flows, many studies have shown that most of the uncertainty lies within the hydrology models (e.g. Giuntoli et al., 2018; Krysanova et al., 2018; Trudel et al., 2017) and removing climate models would have no impact on uncertainty.

The results show that there is no simple answer as to whether or not including hot models in climate change impact studies. In the absence of any computational limitations, we would recommend using as many climate models as possible and study *at posteriori* the impact of including hot models or not. If a selection of a subset of climate models is necessary (e.g. inability

to use a large ensemble due to limited computational capability or costly to run impact models) removing hot models may be a reasonable option. Evaluating climate model fitness for impact studies is a difficult endeavour, and in addition to ECS, additional performance metrics should also carefully be taken into account.

## 5. Conclusion

This study examines the impact of removing a subset of hot climate models on the spread of future projections of streamflow for 3,107 North American catchments. Three streamflow metrics were considered: mean annual streamflow, as well as the mean of the annual maximum and minimum streamflow, over the reference period (1971-2000) and future period (2070-2099).

Hot climate models are determined based on their global equilibrium climate sensitivity (ECS), whereas impact studies typically focus on the local to regional scale. The hot climate models remain among the hottest in our regional evaluation, but they also tend to be among the wettest, potentially leading to a complex hydrological response.

Our research revealed mixed impacts of removing the hot climate models. A decrease in the variability of projected streamflow metrics was generally observed in Canada and Alaska, the southeast US, and the Pacific coast of the US. However, in other regions, removing the hot models resulted in no changes, and in some cases, even increases in the variability of projected flows. This suggests that the hot models are not necessarily hydrological outliers, raising questions about using global performance metrics rather than regional ones for model selection.

The findings of this study emphasize the importance of carefully selecting climate models and the potential risks of including inadequate models in impact studies. In the absence of constraints, it is recommended to use as many climate models as possible in determining impact uncertainty and to assess the impact of subsets of climate models (based on high global equilibrium climate sensitivity or other performance metrics) a posteriori to assess the sensitivity of the impact model to climate model selection. These results highlight the need for further research on climate model fitness and the proper selection of model subsets for impact studies.

**Code and data availability.**

The hydrometeorological data used in this study was obtained from the HYSETS database, which is available at https://doi.org/10.17605/OSF.IO/RPC3W (Arsenault et al., 2020). The CMIP6 GCM model outputs are accessible through

the Earth System Grid Federation Portal at Lawrence Livermore National Laboratory (https://esgf-node.llnl.gov/search/cmip5/). The processed data and the used codes are available via contacting the authors.

## Author Contribution

The experiments were designed by FB, and they were carried out by MRA. The findings were analysed and interpreted by MRA, and FB. The paper was written by MRA and FB, with significant contributions from JLM and RA. JLM and RA also provided editorial feedback on the paper's early draughts.

## Competing interests.

The authors declare that they have no conflict of interest.

## Acknowledgments

This work was supported by the Natural Science and Engineering Research Council (NSERC) of Canada.

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
