# Peer review of "Understanding the Influence of 'Hot' Models in Climate Impact Studies: A Hydrological Perspective"

_Hydrology and Earth System Sciences, 2023_

## Author Comment (AC1)

The authors investigated how the "hot model problem" is critical for the future changes in mean, high and low flows in the North America. I have some minor comments.

We appreciate the time you've taken to read our paper and offer valuable suggestions for enhancement. You will find below a point by point answer to your comments.

L20-23: At the same time, inclusions of hot models lead to critical risks of significant overestimations of climate change impact in some areas.

This is absolutely accurate. The current abstract might seem to place undue emphasis on results concerning catchments with minimal impact. We will ensure that the revised abstract more effectively highlights the potential for overestimating these impacts.

Please add references of ESMs in the table 1. Did you compute ECSs by yourself?

Good point. We will add the references Within Table 1 in the revised version. The ECSs values are normally computed by the various climate modeling centers using a set of controlled runs. Most of the ECS values were taken from the Hausfather et al. (2022) paper. A few others were taken from other sources, typically papers originating from the modelling centers. We will add this information in Table 1 in the revised version of the paper.

Line 56: Shiogama et al. (2022, Nature) constrained future annual precipitation changes. Please see their Fig. 3b. Hot models overestimate annual mean precipitation increases in Alaska, Canada and the West US where your spread of flow changes significantly decline by removing the hot models.

Shiogama, H., Watanabe, M., Kim, H. Emergent constraints on future precipitation changes. Nature 602, 612–616 (2022). https://doi.org/10.1038/s41586-021-04310-8

Line 61: Shiogama et al. (2022, ERL) showed that hot models could cause overestimations of aggregated economic impact of future climate changes.

Shiogama, H., et al. (2022) Uncertainty constraints on economic impact assessments of climate change simulated by an impact emulator. Environ. Res. Lett. 17 124028

https://iopscience.iop.org/article/10.1088/1748-9326/aca68d/meta

Thanks for the references. They are very relevant to our paper as they support and explain some of our findings. We will incorporate the references in both the discussion parts of the revised paper.

L183: How did you define outliers?

The outliers are defined using the default settings in the Matlab *boxplot.m* function. Under this default setting, outliers are defined as having a value larger (smaller) than 1.5 times the interquartile range (Q75-Q25). Using this definition, for normally distributed values, 0.7% of values would be considered

outliers.   Rather than adding this to the revised version, we propose to use a different version of boxplots where the whiskers correspond to the 5[th] and 95[th] quantiles, and values below or above would simply be shown and not called outliers.

L322: High ECS of NESM3 does not necessarily mean greater regional warming (Fig. 3).

Correct.  We will emphasize this in the revised version.

Because the total spread ratio analysis would be not sensitive to the removal of the second largest model, it may be better to show standard deviation ratios in Figs. 12 and 13.

Correct.  We will either substitute the metric or show both Figures if the content information warrants it.

---

## Author Comment (AC2)

Thank you for inviting me to review paper: "The Dilemma of Including Hot Models in Climate Impact Studies: A Hydrological Study" by Asenjan et al. At the outset, I should mention that I am a climate researcher (rather than a hydrologist).

The authors address an issue of maximum concern to climate change research, that at present, the CMIP6 ensembles contain a few models with an especially high climate sensitivity ("hot models"). The concern is that such simulations may project impacts that are especially difficult to adapt to – causing alarm - yet, we may find in the decades ahead that these hot-running ESMs are unrealistic.

The authors focus, in particular, on testing the effect of especially warm ESMs on streamflow. I like this approach because the emphasis is on a major impact and of much concern i.e. flood risk.

We appreciate the reviewer for taking the time to review our paper and provide insightful recommendations for improvement. We are pleased to have the insights of a climate scientist in evaluating our work. In the following section, we have provided a detailed response addressing each of the reviewer's comments.

Based on the Abstract alone as a start, in some ways, I like the catchy title, but using the word Dilemma implies something unresolved. In fact, this paper provides definite findings. Maybe something a bit more factual such as e.g. : "Including Hot Climate Models in estimating Streamflow does not alter the assessment their future variability".

'Dilemma' may indeed be a bit too dramatic. We did not settle on your title suggestion, however, because including 'hot' models does affect variability in some regions, while in most others, it indeed has minimal effects. We have yet to settle on a final revised title, but here are the top candidates so far:

- Evaluating the Inclusion of 'Hot' Models in Climate Impact Studies: Insights from a Hydrological Study
- Assessing the Impact of Incorporating 'Hot' Models in Climate Impact Studies: A Hydrological Viewpoint
- Understanding the Influence of 'Hot' Models in Climate Impact Studies: A Hydrological Perspective

Second, in general and as the world warms, there will be a hydrological intensification for the planet. Hence, it might be expected that hot models generate higher river flows. The authors need to give a good reason for concentrating on variability, rather than mean trends. However, reading further into the manuscript, then mean changes e.g. "Qmean" are considered – maybe reword the Abstract to "mean changes and changes to variability"?

Indeed, our paper focuses on the potential to reduce uncertainty in future changes, as this is a significant issue for decision-makers. A high level of uncertainty can be a deterrent to implementing sound adaptation measures. If this uncertainty could be reduced in a scientifically sound manner, it would be extremely beneficial. However, the intensification of the hydrological cycle does not necessarily mean that higher flows will result, as evapotranspiration could also be significantly affected by the increase in mean temperature. The combined impact of both effects depends on geographical location and primary climate zones, and is not straightforward since, as shown in the paper, 'hot' models tend to also be 'wet.' In light of your comment, we propose to add the following figure, which shows the

actual changes as the ratio of the mean projected changes ('hot' models / normal models). Mean flows are indeed increasing over most of the study domain for 'hot' models, except in the south-west regions, where increased evapotranspiration nullifies potential increases in precipitation.

[Figure]

[Figure]

[Figure]

New Figure XX1: Ratio of mean projected changes: 'hot' divided by normal (not 'hot') models.  Upper graph: Qmean; Middle graph (Qmin), Lower graph (Qmax).  A red color indicates that hot models, on average, have lower flows (mean, minimum, maximum).

It is often asked: "Given the problem of climate change is now emerging strongly in the measurement record, why is it so difficult to constrain future warming estimates?". The main reason for this is the historical record also includes a strong cooling aerosol effect. There are a few papers out there that make this point, and could be worth citing around line 45? In other words, we do not know from present-day measurements if we are in a high sensitivity fast warming world with strong contemporary aerosol cooling, or the opposite. That is the usual reason why we cannot reject outlier ESMs.

The reviewer has raised a very good point. We will incorporate this point, along with appropriate citations, into the revised version of the paper.

The description of the methods feels a bit too short in places, which is a shame because the authors have worked especially hard to entrain data, bias-correct a hydrological model against ERA5, bias-correct the ESMS. The test data is especially comprehensive (~14K sites). Would a schematic work that can capture some of this activity?

We appreciate the reviewer's comments. It's always a fine line between having too little vs too much methodological information.  In light of the comment, we will include a schematic to better describe the modeling procedure of this study, and we will add extra details to the methods section for a more comprehensive representation of our methods.

I had not heard of the KGE metric / statistic before, and where it is introduced, there is no citation. Can a reference be provided, and maybe a short sentence that explains its basic features? Presumably, the statistic allows an assessment of which hydrological models are performing best (based on aspects of variability in timeseries?).

Considering your background, it's probably not surprising that the KGE metric, which is rooted in hydrological science, is not familiar to you. Given the multidisciplinary nature of HESS, we will expand the text to provide additional details on this metric, along with references.

When calibrating and assessing hydrological models, performance criteria are generally used to quantify the degree of similarity between observed and simulated discharges. The Nash-Sutcliffe Efficiency (NSE, Nash & Sutcliffe, 1970), a normalized RMSE metric, has long been the standard for assessing model efficiency. However, it has a number of drawbacks, including those affecting bias and correlation, that have been addressed by the Kling-Gupta Efficiency metric (KGE) (Gupta et al., 2009). KGE has gained popularity over the past 10 years and is gradually replacing the NSE in literature dealing with the calibration of hydrological models. The KGE metric directly combines the bias, ratio of variance, and correlation into a single metric.

$$KGE = 1 - \sqrt{(r-1)^2 + (\alpha - 1)^2 + (\beta - 1)^2}$$

where $r$ is the linear correlation between observations and simulations, $\alpha$ a measure of the flow variability error, and $\beta$ a bias term

$$KGE = 1 - \sqrt{(r-1)^2 + (\frac{\sigma_{sim}}{\sigma_{obs}} - 1)^2 + (\frac{\mu_{sim}}{\mu_{obs}} - 1)^2}$$

where $\sigma_{obs}$ is the standard deviation in observations, $\sigma_{sim}$ the standard deviation in simulations, $\mu_{sim}$ the simulation mean, and $\mu_{obs}$ the observation mean.

Please also check the order of presentation. Figure 1 is cited in the text (line 84) before the modelling is described in full (around line 124).

We will take this point into account in the revised version of the manuscript and correct the order of the presentation.

There could be improvement of the manuscript around page 7 (lines 135-158, and Figure 2). Can it be confirmed that Figure 2 is a generic plot, referring to any metric of interest? Line 143 "Figure 2 present the three dispersion metrics…. streamflow". This is too vague – does the "y" axis of Figure 2 related to Qmean, Qmax or Qmin? Apologies if I am missing something obvious.

Figure 2 is indeed a generic plot referring to all the metrics considered in the manuscript. The "y" axis of Figure 2 represents the Streamflow metrics, e.g. Qmean, Qmax or Qmin, while the X axis represents the time. we will improve the explanation in the revised version to avoid any confusion. In addition, we will make minor modifications to the Figure.

In results, line 186, there is reference to ECS. However, neither here, or in the caption to Figure 3 (or caption to the Table) is a reference to the source of the ECS values. For Figure 3, you could potentially use some sort of marker to differentiate the five warmest models on the left. For instance, a horizontal arrow under the first five models, with the words "warmest models".

Correct, this point was also mentioned by the first reviewer, and we will include the references to the source of ECS values in the revised version.

We will also edit figure 3 to distinguish the hot models from the rest.

In places I thought the captions to diagrams and tables were very short. Obviously, captions cannot repeat everything that is in the paper text, but a little more information in places might help the reader (this is important if, upon publication, people extract diagrams and captions to place in powerpoints for talks).

We acknowledge that in some instances, the captions may rely too heavily on the description provided in the main text. In the revised version of the paper, we will ensure that we include more details in the figure and table captions. This will allow each figure or table to be understood independently

This might be something for the HESS editor to advise on, but in my view, the main plot of the paper is Figure 5. This presents very clearly the geographical effect of removal of models with a high ECS. After that, there are many helpful diagrams, but I am not convinced that all are needed. Or some sort of multi-panel plot might help, e.g. merging the box plots. The authors could consider placing some of the additional information and plots after that in an SI – e.g. Figure 12.

We appreciate the reviewer's suggestion. We agree that relocating some figures to the supporting information could help emphasize the main points of our paper. However, we also believe that retaining some figures, such as those beyond figure 5, is necessary to effectively communicate the crux of our research. For instance, figure 5 displays the total spread ratio, a measure of the data range that is heavily influenced by outliers. Conversely, figure 9 illustrates the standard deviation ratio, a measure of data dispersion. We recognize the similarity in the patterns shown in Figures 7 and 9, so much so that we could move either Figure 7 or Figure 9 to the supporting information while retaining the discussion in the revised manuscript.

We will also merge the boxplots with their corresponding figures (Figure 5 with 6, Figure 7 with 8, Figure 9 with 10), and move Figures 12 and 13 to the supporting information.

One small frustration with this manuscript is that once variables are declared, and given abbreviated names, often such names are not then used systematically through the paper. This is especially noticeable when presenting keynote diagrams, such as Figure 5. Hence, the caption for Figure 5 should read something like: "Total spread ratio, TS_nd, for Q_mean….."

Good point. In the revised version of the paper, we will ensure that we use the abbreviated names of each variable systematically.

Would the authors like to comment on the spatial cohesion in plots such as Figure 5. This diagram in particular seems to show that it is the higher latitudes that will see the bigger effects if the high ECS models are correct? Certainly the largest levels of warming will be seen towards to poles, which is a generic result for ESMs. That is the ESMs with the largest ECS which show the most pronounced warming poleward. This might also be true for precipitation?

To address this valid point, we propose adding a figure showing the mean climate change ratio for $\Delta T$ and $\Delta P/P$. As discussed earlier, the ratio represents the mean projected changes ('hot' models to normal models). These figures will assist in interpreting some of our other figures. This new figure reveals intriguing patterns, particularly for precipitation. The temperature ratio is relatively consistent, with the additional warming projected by the 'hot' models always exceeding 1 and showing less spatial dependency compared to precipitation. For precipitation, strong spatial patterns emerge. The 'hot' models predict considerably higher precipitation over central North America, while they project lower precipitation over the West coast of the USA.

[Figure]

New Figure XX2: Mean ΔT (left-hand side) and ΔP/P (right-hand side) ratios (hot models to normal models). For ΔT, a red color indicates that hot models, on average, are warmer than their normal (non-hot) counterparts. For ΔP/P, a blue color shows that hot models are wetter than their normal (non-hot) counterparts. ΔT represents the difference between future and historical temperatures.

It is noted that the reference list is balanced and comprehensive. I also like the Introduction, and the references cited where attempts have been made (either via emergent constraints or historical data) to constrain the bounds of Equilibrium Climate Sensitivity (ECS) bounds. This paper contains a wealth of important information, addresses an important problem of how to deal with ESM outliers and in the important context of such model impacts on streamflow. I am very happy to see any other revised paper version.

We appreciate the reviewer's positive feedback and insightful comments. We will ensure that the revised version of the paper addresses the identified shortcomings and points raised.

---

## Author Response (AR1)

The authors investigated how the "hot model problem" is critical for the future changes in mean, high and low flows in the North America. I have some minor comments.

We appreciate the time you've taken to read our paper and offer valuable suggestions for enhancement. You will find below a point by point answer to your comments.

L20-23: At the same time, inclusions of hot models lead to critical risks of significant overestimations of climate change impact in some areas.

This is absolutely accurate. We have revised the abstract to effectively highlight the potential for overestimating climate change impacts.

We have added a sentence at the end of the first paragraph of the abstract to better emphasize this. The first paragraph of the abstract now reads like this (with the added sentence in **bold**):

"Efficient adaptation strategies to climate change require estimating future impacts and the uncertainty surrounding this estimation. Over- or under-estimating future uncertainty may lead to maladaptation. Hydrological impact studies typically use a top-down approach in which multiple climate models are used to assess the uncertainty related to climate model structure and climate sensitivity. Despite ongoing debate, impact modelers have typically embraced the concept of "model democracy" in which each climate model is considered equally fit. The newer CMIP6 simulations, with several models showing a climate sensitivity larger than that of CMIP5 and larger than the likely range based on past climate information and understanding of planetary physics, have reignited the model democracy debate. Some have suggested that hot models be removed from impact studies to avoid skewing impact results toward unlikely futures. **Indeed, the inclusion of these models in impact studies carries a significant risk of overestimating the impact of climate change.**"

Please add references of ESMs in the table 1. Did you compute ECSs by yourself?

Good point. We have added the references to the table 1 in the revised version. The ECSs values are normally computed by the various climate modeling centers using a set of controlled runs. The ECS values were taken from (Tokarska et al., 2020) and (Hausfather et al., 2022) which have been cited in Table 1.

Line 56: Shiogama et al. (2022, Nature) constrained future annual precipitation changes. Please see their Fig. 3b. Hot models overestimate annual mean precipitation increases in Alaska, Canada and the West US where your spread of flow changes significantly decline by removing the hot models.

Shiogama, H., Watanabe, M., Kim, H. Emergent constraints on future precipitation changes. Nature 602, 612–616 (2022). https://doi.org/10.1038/s41586-021-04310-8

Line 61: Shiogama et al. (2022, ERL) showed that hot models could cause overestimations of aggregated economic impact of future climate changes.

Shiogama, H., et al. (2022) Uncertainty constraints on economic impact assessments of climate change simulated by an impact emulator. Environ. Res. Lett. 17 124028

https://iopscience.iop.org/article/10.1088/1748-9326/aca68d/meta

Thanks for the references. They are very relevant to our paper as they support and explain some of our findings. We have included these references in the introduction. we have also used these references in the discussion part of our paper. (the second one in introduction line 73 and the first one in discussion line 367 in the track-change version of the manuscript.)

L183: How did you define outliers?

The outliers are defined using the default settings in the Matlab *boxplot.m* function. Under this default setting, outliers are defined as having a value larger (smaller) than 1.5 times the interquartile range (Q75-Q25). Using this definition, for normally distributed values, 0.7% of values would be considered outliers. Rather than adding this to the revised version, we have modified the boxplots where the whiskers correspond to the 2.5$^{th}$ and 97.5$^{th}$ quantiles, and values below or above have been simply shown and not called outliers.

L322: High ECS of NESM3 does not necessarily mean greater regional warming (Fig. 3).

Correct. We have put more emphasis on this point in the revised version. For example, see lines 417 to 421 in the track-change version of the manuscript.

"Removing CanESM5 leads to a clear reduction of total spread in Alaska and Yukon (for $Q_{mean}$ and $Q_{min}$) and in the Southeast USA for $Q_{max}$, indicating that CanESM5 is an outlier in these regions. Conversely, removing NESM3 does not result in significant decreases in spread over most of the study domain, as the high ECS value of NESM3 does not automatically translate into a correspondingly higher level of regional warming (see also fig. 4), demonstrating that it is not an outlier in most regions. This underscores the strong regional differences among globally identified hot models."

Because the total spread ratio analysis would be not sensitive to the removal of the second largest model, it may be better to show standard deviation ratios in Figs. 12 and 13.

Correct. We have moved the figs 12 and 13 to the supplementary information as we see that the main point of the paper has already been emphasised enough.

Added references

Hausfather, Z., Marvel, K., Schmidt, G. A., Nielsen-Gammon, J. W., & Zelinka, M. (2022). Climate simulations: Recognize the 'hot model' problem. *Nature*, *605*(7908), 26–29. https://doi.org/10.1038/d41586-022-01192-2

Tokarska, K. B., Stolpe, M. B., Sippel, S., Fischer, E. M., Smith, C. J., Lehner, F., & Knutti, R. (2020). Past warming trend constrains future warming in CMIP6 models. *Science Advances*, *6*(12), eaaz9549. https://doi.org/10.1126/sciadv.aaz9549

Forster, P. M., Andrews, T., Good, P., Gregory, J. M., Jackson, L. S., & Zelinka, M.: Evaluating adjusted forcing and model spread for historical and future scenarios in the CMIP5 generation of climate models, Journal of Geophysical Research: Atmospheres, 118(3), 1139–1150, https://doi.org/10.1002/jgrd.50174, 2013.

Knoben, W. J. M., Freer, J. E., & Woods, R. A.: Technical note: Inherent benchmark or not? Comparing Nash-Sutcliffe and Kling-Gupta efficiency scores, Catchment hydrology/Modelling approaches, https://doi.org/10.5194/hess-2019-327, 2019.

Shiogama, H., Takakura, J., & Takahashi, K.: Uncertainty constraints on economic impact assessments of climate change simulated by an impact emulator, Environmental Research Letters, 17(12), 124028, https://doi.org/10.1088/1748-9326/aca68d, 2022.

Shiogama, H., Watanabe, M., Kim, H., & Hirota, N.: Emergent constraints on future precipitation changes. Nature, 602(7898), 612–616. https://doi.org/10.1038/s41586-021-04310-8, 2022.

Smith, C. J., Harris, G. R., Palmer, M. D., Bellouin, N., Collins, W., Myhre, G., Schulz, M., Golaz, J. -C., Ringer, M., Storelvmo, T., & Forster, P. M.: Energy Budget Constraints on the Time History of Aerosol Forcing and Climate Sensitivity, Journal of Geophysical Research: Atmospheres, 126(13), https://doi.org/10.1029/2020JD033622. 2021.

Thank you for inviting me to review paper: "The Dilemma of Including Hot Models in Climate Impact Studies: A Hydrological Study" by Asenjan et al. At the outset, I should mention that I am a climate researcher (rather than a hydrologist).

The authors address an issue of maximum concern to climate change research, that at present, the CMIP6 ensembles contain a few models with an especially high climate sensitivity ("hot models"). The concern is that such simulations may project impacts that are especially difficult to adapt to – causing alarm - yet, we may find in the decades ahead that these hot-running ESMs are unrealistic.

The authors focus, in particular, on testing the effect of especially warm ESMs on streamflow. I like this approach because the emphasis is on a major impact and of much concern i.e. flood risk.

We appreciate the reviewer for taking the time to review our paper and provide insightful recommendations for improvement. We are pleased to have the insights of a climate scientist in evaluating our work. In the following section, we have provided a detailed response addressing each of the reviewer's comments.

Based on the Abstract alone as a start, in some ways, I like the catchy title, but using the word Dilemma implies something unresolved. In fact, this paper provides definite findings. Maybe something a bit more factual such as e.g. : "Including Hot Climate Models in estimating Streamflow does not alter the assessment their future variability".

'Dilemma' may indeed be a bit too dramatic. We did not settle on your title suggestion, however, because including 'hot' models does affect variability in some regions, while in most others, it indeed has minimal effects. We have changed the title of our paper to "Understanding the Influence of 'Hot' Models in Climate Impact Studies: A Hydrological Perspective".

Second, in general and as the world warms, there will be a hydrological intensification for the planet. Hence, it might be expected that hot models generate higher river flows. The authors need to give a good reason for concentrating on variability, rather than mean trends. However, reading further into the manuscript, then mean changes e.g. "Qmean" are considered – maybe reword the Abstract to "mean changes and changes to variability"?

Indeed, our paper focuses on the potential to reduce uncertainty in future changes, as this is a significant issue for decision-makers. A high level of uncertainty can be a deterrent to implementing sound adaptation measures. If this uncertainty could be reduced in a scientifically sound manner, it would be extremely beneficial. However, the intensification of the hydrological cycle does not necessarily mean that higher flows will result, as evapotranspiration could also be significantly affected by the increase in mean temperature. The combined impact of both effects depends on geographical location and primary climate zones, and is not straightforward since, as shown in the paper, 'hot' models tend to also be 'wet.' In light of your comment, we have added the following figures, which show the mean $\Delta T$ and $\Delta P/P$ ratios (hot models vs the others) as the new Figure 5 as well as actual streamflow changes as the ratio of the mean projected changes ('hot' models / normal models) in the new Figure 6. Figure 5 reveals intriguing patterns, particularly for precipitation. The temperature ratio is relatively consistent, with the additional warming projected by the 'hot' models always exceeding 1 and showing less spatial dependency compared to precipitation. For precipitation, strong spatial patterns emerge.

The 'hot' models predict considerably higher precipitation over central North America, while they project lower precipitation over the West coast of the USA.

Figure 6 shows that mean flows are indeed increasing over most of the study domain for 'hot' models, except in the southwest regions, where increased evapotranspiration nullifies potential increases in precipitation.

[Figure]

*Figure 5. Mean ΔT (left-hand side) and ΔP/P (right-hand side) ratios (hot models to normal models). For ΔT, a red color indicates that hot models, on average, are warmer than their normal (non-hot) counterparts. For ΔP/P, a blue color shows that hot models are wetter than their normal (non-hot) counterparts. The graphs represent the differences computed between the future and reference periods.*

[Figure]

*Figure 6: Ratio of mean projected changes: 'hot' divided by normal models.  a): Qmean; b) Qmin; c): Qmax). A blue color shows that hot project larger streamflows than their normal (non-hot) counterparts.*

It is often asked: "Given the problem of climate change is now emerging strongly in the measurement record, why is it so difficult to constrain future warming estimates?". The main reason for this is the historical record also includes a strong cooling aerosol effect. There are a few papers out there that make this point, and could be worth citing around line 45? In other words, we do not know from present-day measurements if we are in a high sensitivity fast warming world with strong contemporary aerosol cooling, or the opposite. That is the usual reason why we cannot reject outlier ESMs.

The reviewer has raised a very good point. We have added the following to the manuscript in lines 63 in the track-change version of the manuscript:

"It should be noted that the uncertainty surrounding the cooling impact of aerosols (both direct and indirect) on radiative forcing poses challenges in constraining future warming estimates (Bellouin et al., 2020; Forster et al., 2013; Smith et al., 2021). In essence, the current measurements do not provide a clear understanding of whether we are in a scenario of high sensitivity, fast-warming, accompanied by strong contemporary aerosol cooling, or if the opposite is true."

The description of the methods feels a bit too short in places, which is a shame because the authors have worked especially hard to entrain data, bias-correct a hydrological model against ERA5, bias-correct the ESMS. The test data is especially comprehensive (~14K sites). Would a schematic work that can capture some of this activity?

We appreciate the reviewer's comments. It's always a fine line between having too little vs too much methodological information.  In light of the comment, we have included the following details and schematic to better describe the modeling procedure of this study.

"Figure 1 illustrates the methodological framework for each study catchment. Precipitation and temperature data are first extracted from 19 CMIP6 climate models under the SSP8.5 scenario for both the reference and future periods. Using precipitation and temperature from the ERA5 reanalysis over the reference period, climate data is then bias-corrected using the MBCn method. These bias-corrected climate scenarios are subsequently employed as inputs for a calibrated hydrological model to compute streamflows. These computed streamflows are then used to examine the impact of including (or not including) 'hot' models in the impact study, using a set of defined metrics. Further details are provided below.

[Figure]

Figure 1. Methodological framework performed for each of the study catchments."

I had not heard of the KGE metric / statistic before, and where it is introduced, there is no citation. Can a reference be provided, and maybe a short sentence that explains its basic features? Presumably, the statistic allows an assessment of which hydrological models are performing best (based on aspects of variability in timeseries?).

Considering your background, it's probably not surprising that the KGE metric, which is rooted in hydrological science, is not familiar to you. Given the multidisciplinary nature of HESS, we have expanded the text to provide additional details on this metric, along with references.

Please also check the order of presentation. Figure 1 is cited in the text (line 84) before the modelling is described in full (around line 124).

We have corrected the order of the presentation.

There could be improvement of the manuscript around page 7 (lines 135-158, and Figure 2). Can it be confirmed that Figure 2 is a generic plot, referring to any metric of interest? Line 143 "Figure 2 present the three dispersion metrics…. streamflow". This is too vague – does the "y" axis of Figure 2 related to Qmean, Qmax or Qmin? Apologies if I am missing something obvious.

Figure 2 (figure 3 in the revised version) is indeed a generic plot referring to all the metrics considered in the manuscript. The "y" axis of Figure 2 (figure 3 after revision) represents the Streamflow metrics, e.g. Qmean, Qmax or Qmin, while the X axis represents the time. we have added additional information in the Figure caption to better outline this.

In results, line 186, there is reference to ECS. However, neither here, or in the caption to Figure 3 (or caption to the Table) is a reference to the source of the ECS values. For Figure 3, you could potentially use some sort of marker to differentiate the five warmest models on the left. For instance, a horizontal arrow under the first five models, with the words "warmest models".

Correct, this point was also mentioned by the first reviewer. We have added the references to the table 1 in the revised version. The ECS values are typically calculated by different climate modeling centers through a series of controlled simulations. The ECS values referenced in Table 1 were extracted from (Tokarska et al., 2020) and (Hausfather et al., 2022).

We have also merged figures 3 and 4, and discriminated between "hot" models and the other models with a red dashed line and the phrase "warmest models"

In places I thought the captions to diagrams and tables were very short. Obviously, captions cannot repeat everything that is in the paper text, but a little more information in places might help the reader (this is important if, upon publication, people extract diagrams and captions to place in powerpoints for talks).

We acknowledge that in some instances, the captions may rely too heavily on the description provided in the main text. we have included more details in the figure and table captions. This will allow each figure or table to be understood independently

This might be something for the HESS editor to advise on, but in my view, the main plot of the paper is Figure 5. This presents very clearly the geographical effect of removal of models with a high ECS. After that, there are many helpful diagrams, but I am not convinced that all are needed. Or some sort of multi-panel plot might help, e.g. merging the box plots. The authors could consider placing some of the additional information and plots after that in an SI – e.g. Figure 12.

We appreciate the reviewer's suggestion. We agree that relocating some figures to the supporting information could help emphasize the main points of our paper. To this point we have moved figure 7

and 8 to the supporting information and merged figured 5 and 6 and figures 9 and 10, which are figures 7 and 8 in the revised version. We have also moved figures 12 and 13 to the supporting information.

However, we also believe that retaining some figures, such as those beyond figure 7 (in the revised version), is necessary to effectively communicate the crux of our research. For instance, figure 7 (in the revised version) displays the total spread ratio, a measure of the data range that is heavily influenced by outliers. Conversely, figure 8 (in the revised version) illustrates the standard deviation ratio, a measure of data dispersion.

One small frustration with this manuscript is that once variables are declared, and given abbreviated names, often such names are not then used systematically through the paper. This is especially noticeable when presenting keynote diagrams, such as Figure 5. Hence, the caption for Figure 5 should read something like: "Total spread ratio, TS_nd, for Q_mean….."

Good point. we have revised the paper to systematically use the abbreviated names.

Would the authors like to comment on the spatial cohesion in plots such as Figure 5. This diagram in particular seems to show that it is the higher latitudes that will see the bigger effects if the high ECS models are correct? Certainly the largest levels of warming will be seen towards to poles, which is a generic result for ESMs. That is the ESMs with the largest ECS which show the most pronounced warming poleward. This might also be true for precipitation?

We believe we have addressed this issue with the addition of two new Figures (Figure 5 and 6) as per your second general comment. These figures help with the interpretation of our other figures.

It is noted that the reference list is balanced and comprehensive. I also like the Introduction, and the references cited where attempts have been made (either via emergent constraints or historical data) to constrain the bounds of Equilibrium Climate Sensitivity (ECS) bounds. This paper contains a wealth of important information, addresses an important problem of how to deal with ESM outliers and in the important context of such model impacts on streamflow. I am very happy to see any other revised paper version.

We appreciate the reviewer's positive feedback and insightful comments. We have ensured that the revised version of the paper addresses the identified shortcomings and points raised.

---

## Author Response (AR2)

Please accept my apologies for the lengthy time to return a re-review of the paper now titled "Understanding the Influence of Hot Models in Climate Impact Studies: A Hydrological Perspective".

I can see the authors have taken seriously both my suggestions and those of the other reviewer.

We appreciate your time and effort in reviewing our work and your valuable feedback, which has helped us enhance the quality of our manuscript.

I am really pleased that the authors have removed the word "Dilemma", and please check for any other dramatic wording in the manuscript. Although we may have views on the dangers of global warming, it is important for researchers to write in a way that is impartial.

we have reviewed the entire document to identify and replace any other instances of dramatic language or wording that may have crept in.

The extra diagrams are appreciated and especially Figure 5. In general, models with high ECS are expected to be wetter for the same levels of atmospheric GHGs. The disaggregation revealing Californian drying (Figure 5b) for the warmest models is fascinating. Hence, I like the additional sentences starting with "However, the west coast of the US…". Such spatial heterogeneity has strong policy implications for the US, as reflected in the new streamflow Figure 6. Not for this manuscript, but maybe the authors might like to consider a follow-on analysis, picking apart in more detail what is projected for California – potentially in the context of fire risk.

We are delighted to hear that you found the additional diagrams, especially Figure 5, valuable and that the information provided in our manuscript has piqued your interest.

Your observation regarding the effects of high ECS models on regional climate, specifically the phenomenon of Californian drying as depicted in Figure 5b, is indeed a compelling aspect of our findings. We agree that the spatial heterogeneity highlighted in our study has important policy implications, particularly for the United States.

Regarding your suggestion for a follow-on analysis focusing on California and its relation to fire risk, we find this idea intriguing and highly relevant. Exploring the projected climate changes in California with a focus on fire risk would certainly be a valuable extension of our current research. We will consider this suggestion for future work and appreciate your input on this potential research avenue.

This paper is now ready for publication. I would advise the authors to make one final run-through of the figures to check for any very last-minute formatting issues. For instance, maintain the standard aspect ratio for maps – Figure 5 looks as if the longitudinal axis has been suppressed. Some of the "box-and-whiskers" plots look like they have substantial white space and could be reduced in size?

Thank you for your positive feedback and your recommendation that our paper is ready for publication. We greatly appreciate your careful review and valuable suggestions.

We have made the necessary adjustments to Figure 5 to ensure it maintains the standard aspect ratio. Additionally, we have reviewed the boxplots, particularly Figure 9, and have eliminated substantial white space and adjusted their size.